# Indocyanine green fluorescence lymphography: An exploratory study of superficial lymphatic territories in the head and hind limbs of 33 cat cadavers

Alessandra Ubiali[1]*, Elisa Maria Gariboldi[1], Luigi Auletta[1], Alessia Di Giancamillo[2], Silvia Clotilde Bianca Modina[1], Roberta Ferrari[1], Filippo Tagliasacchi[1], Valeria Martini[1], Damiano Stefanello[1]

1 Department of Veterinary Medicine and Animal Sciences, University of Milan, Lodi, Italy, 2 Department of Biomedical Sciences, University of Milan, Milano, Italy

☯ These authors contributed equally to this work.
* alessandra.ubiali@unimi.it

## Abstract

To date, animal models for lymphographic studies mainly focused on dog, while lymphography is rarely reported in cats, and even less involving cutaneous lymphatic territories. This study aims to assess the feasibility of cutaneous lymphography using indocyanine green (ICG) fluorescence in cat cadavers and describe predictable lymphatic pathways from cutaneous regions of head and hind limb anatomical districts. Frozen or refrigerated cadavers of adult cats that died for causes unrelated to the study were included. Twenty cutaneous regions (6 from the head; 14 from the hind limb) were selected using easily assessable anatomical landmarks, and expected draining lymphocentrums were presumed based on canine studies since there is no similar information for cats. For each lymphography, a single selected cutaneous region per anatomical district was assessed. After intradermal ICG injections, lymphatic drainage was favored by massage and/or flexion-extension movements. For each lymphography, all expected and detected lymphocentrums were dissected, and lymph nodes extirpated. Variables regarding cadavers and lymphography characteristics were assessed. ICG-lymphography was repeated in 33 cadavers. Out of the 99 selected cutaneous regions available, 15 were excluded following inclusion criteria, therefore lymphographies were performed for a total of 84 selected cutaneous regions (26 from the head and 58 from the hind limbs). A success was recorded in 63/84 (75%) lymphographies, with a median migration time of 8 (1–30) minutes. The ICG drained to the expected lymphocentrum in 28/63 (44%) lymphographies, and to other ones in 35/84 (56%). ICG-lymphography is feasible in cat cadavers, regardless of technique or cadaver characteristics. The observed difference in lymphatic drainage (56% to unexpected lymphocentrums) highlights the importance of specifically mapping lymphatic territories in cats. ICG-lymphography demonstrated as

**Data availability statement:** All relevant data are within the manuscript and its Supporting Information files.

**Funding:** The author(s) received no specific funding for this work.

**Competing interests:** The authors have declared that no competing interests exist.

an effective technique and could be used to improve knowledge of feline lymphatic physiology. Further studies may provide a more complete understanding of superficial lymphatic territories in cats.

## Introduction

In the last decades, lymphography in cadavers has been performed in various animal species as anatomic and translational models for human lymphedema and lymphatic alterations [1–4]. Numerous techniques have been described for assessing superficial lymphatic drainage in human and animal cadavers [1–7]. Among them, near-infrared fluorescence lymphography with indocyanine green (NIRF-ICG) has become the most used technology [7–11].

Data obtained on canine cadavers [7] has been applied in several clinical studies to predict the expected regional draining lymphocentrum (LC) and compare it to the sentinel LC effectively identified [12–16].

Conversely, only few descriptive anatomical studies on the feline lymphatic system are available [17,18]. Furthermore, lymphographic studies in cats are outdated [5,6] or focused only on the mammary gland, either *in vivo* or in cadavers [19–21]. Hence, no lymphographic studies detailing the cutaneous lymphatic territories have been performed in cats. As a result, clinical studies involving lymph node (LN) assessment in cats rely only on what was reported in dogs [22], leading to potential inaccuracies.

Recent advancements in techniques for mapping and removing the sentinel lymph node in dogs have demonstrated that the superficial lymphatic network of the head and hind limb regions is challenging to identify due to its anatomical complexity [12,23,7], potentially leading to technique failures. With this in mind, the present study aims to assess the feasibility of NIRF-ICG lymphography in feline cadavers and to map the superficial lymphatic pathways from selected cutaneous regions of the head and hind limbs, with the goal of detailing potential "lymphosomes." Additionally, the study will explore differences in lymphographic patterns between these two anatomical regions.

## Materials and methods

This explorative study included cadavers of both client-owned and unowned free-roaming adult cats that died spontaneously or were euthanized for causes unrelated to the study. Written consent was obtained from the owners (owned cats) or the public veterinary officer (unowned free-roaming cats) for the scientific use of cadavers.

All procedures were performed at the Department of Veterinary Medicine and Animal Sciences of the University of Milano.

### Case selection

Adult feline cadavers with non-palpable, normal-sized (not clinically enlarged, compared to the contralateral) superficial LNs were included and underwent NIRF-ICG

lymphography of selected cutaneous regions belonging to different anatomical districts: head, right hind limb, and left hind limb. The cadavers were defined as "adults" if no deciduous teeth were observed; otherwise, they were defined as "juveniles" and consequently not included. Cadavers were also excluded when they presented evident and severe cadaveric alterations (i.e., extensive areas of hypostasis, cadaveric hemolysis, diffuse tissue autolysis, putrefaction signs like cadaveric emphysema, and tissue colliquation). In addition, each anatomical district was also individually evaluated and was excluded in case of macroscopical alterations such as neoplastic or cutaneous lesions as a consequence of local cadaveric alteration or due to wounds or scars.

The BCS (Body Condition Score) was assigned on a five-point scale and defined as: emaciated, thin, normal, moderately overweight, or obese [24]. For statistical purposes, cadavers were categorized into two classes considering BCS: low BCS (pooling emaciated and thin) and normal-high BCS (pooling normal, moderately overweight and obese) [25]. Moreover, cats were categorized based on craniofacial type into brachycephalic and non-brachycephalic [26,27].

## Cadaver preparation for NIRF-ICG lymphography

Cadavers were stored within 1 hour from death at +4°C if lymphography was planned within 24 hours; otherwise, they were stored at −20°C. Based on storage conditions, cadavers were categorized as "refrigerated" if stored at +4°C or "frozen" if stored at −20°C. Frozen cadavers were thawed and kept at +4°C until completely defrosted, i.e., when flexion and extension of the joints could be performed without effort, before the preparation for lymphography. Regardless of the storage conditions, cadavers were kept for one hour at room temperature before starting the NIRF-ICG lymphography.

Immediately before NIRF-ICG lymphography, the anatomical districts were widely clipped. The head was clipped from the top of the nose to the scapular spine on both sides, and the hind limbs were clipped from digits to the iliac crest and ischiatic tuberosity, medially including the inguinal region. Cadavers were manipulated carefully during the preparation, ensuring that the skin remained macroscopically undamaged during trichotomy and therefore superficial lymphatic vessels were not transected.

## Definition of the cutaneous regions within the anatomical districts

Within the anatomical districts, a total of 20 cutaneous regions were predetermined: 6 for the head and 14 for the hind limb (Tables 1 and 2). Since the head is an unpaired anatomical district, all median regions were excluded to avoid possible bilateral migrations. For paired bilateral regions, margins were drawn at least 0.5 cm apart from the midline. The borders of

**Table 1. Selected cutaneous regions in the head anatomical district.**

| Selected cutaneous region | Anatomical landmarks | Predictability (predictable/ unpredictable) | expected LC |
|---|---|---|---|
| **Auricular** | Semi-circumferential region delimiting the caudal part of the ear cartilage, within 1 cm from the edge. Pinna was not included. | PREDICTABLE | Superficial cervical |
| **Rostral mandibular** | Cutaneous portion of the inferior lip from tooth 302 (or 402) to tooth 307 (or 407), ventrally delimited by the ventral ridge of the mandibular bone. | PREDICTABLE | Mandibular |
| **Aboral mandibular** | Cutaneous portion of the inferior lip from tooth 307 (or 407) to the labial commissure, ventrally delimited by the ventral ridge of the of mandibular bone. | PREDICTABLE | Mandibular |
| **Rostral maxillary** | Cutaneous portion of the superior lip from tooth 102 (or 202) to tooth 107 (or 207), dorsally delimited by the alar cartilage and medial canthus of the eye. | PREDICTABLE | Mandibular |
| **Aboral maxillary** | Cutaneous portion of the superior lip from tooth 107 (or 207) to the labial commissure, dorsally delimited by the border between maxillary and zygomatic bones. | PREDICTABLE | Mandibular |
| **Temporal-zygomatic** | Triangle of skin delimited by the lateral canthus of the eye, proximal and distal edges of the cranial aspect of ear base. | PREDICTABLE | Parotid |

**Table 2. Selected cutaneous regions in the hind limb anatomical district.**

| Selected cutaneous region | Anatomical landmarks | Predictability (predictable/ unpredictable) | expected LC |
|---|---|---|---|
| **Lateral thigh – cranial** | Region delimited by the lateral trochlear ridge of femur, lateral condyle of femur, tuberosity of greater trochanter, and tuber coxae. | **UNPREDICTABLE** | Superficial inguinal Medial Iliac |
| **Lateral thigh – caudal** | Region delimited by the lateral condyle of femur, popliteal fossa, ischiatic tuberosity, and tuberosity of greater trochanter. | **UNPREDICTABLE** | Superficial inguinal Medial Iliac Sacral |
| **Medial thigh – cranial** | Region delimited by the medial trochlear ridge of femur, medial condyle of femur, laterally delimited femur and fold of flank until the end of abdominal wall proximally. | PREDICTABLE | Superficial inguinal |
| **Medial thigh – caudal** | Region delimited by the medial condyle of femur, laterally delimited by femur and caudal aspect of thigh, to the pubis. | PREDICTABLE | Superficial inguinal |
| **Lateral genicular** | Region delimited by the lateral trochlear ridge of femur, popliteal fossa, and lateral tibial condyle tuberosity. | PREDICTABLE | Superficial inguinal |
| **Medial genicular** | Region delimited by the medial trochlear ridge of femur, popliteal fossa, and medial tibial condyle tuberosity. | PREDICTABLE | Superficial inguinal |
| **Lateral crural** | Region delimited by the lateral tibial condyle tuberosity, popliteal fossa, fibula distal tuberosity, and calcaneal tuberosity. | **UNPREDICTABLE** | Superficial inguinal Popliteal |
| **Medial crural** | Region delimited by the medial tibial condyle tuberosity, popliteal fossa, tibial distal tuberosity, and calcaneal tuberosity. | PREDICTABLE | Superficial inguinal |
| **Lateral tarsal** | Region delimited by the fibular lateral malleolus, calcaneal tuberosity, and base of V metatarsal bone. | **UNPREDICTABLE** | Superficial inguinal Popliteal |
| **Medial tarsal** | Region delimited by the tibial medial malleolus, calcaneal tuberosity, base of II metatarsal bone. | PREDICTABLE | Superficial inguinal |
| **Dorsal metatarsal** | Dorsal aspect of the region delimited proximally by the base of V and II metatarsal bones, and distally by metatarsophalangeal joints. | PREDICTABLE | Popliteal |
| **Plantar metatarsal** | Plantar aspect of the region delimited proximally by the base of V and II metatarsal bones and distally by metatarsophalangeal joints. | PREDICTABLE | Popliteal |
| **Dorsal phalangeal** | Dorsal aspect of the region delimited by the metatarsophalangeal joints proximally and base of the claws distally. | PREDICTABLE | Popliteal |
| **Plantar phalangeal** | Plantar aspect of the region delimited by the metatarsophalangeal joints proximally and base of claws distally. | PREDICTABLE | Popliteal |

each selected cutaneous region were defined based on readily identifiable and unambiguous anatomical landmarks from already described topographic regions [28]. Each selected cutaneous region and the corresponding anatomical landmarks are described in Tables 1 and 2 and Figs 1 and 2. Each selected cutaneous region was drawn with a skin marker (Fig 3).

The expected draining LC was recorded for each selected cutaneous region. Since the authors are not aware of any previous study that has identified LC based on specific lymphatic territories in cats, an LC was classified as "predictable" if the defined region was expected to drain to a single LC, or "unpredictable" if the region corresponded to more than one lymphosome based on canine lymphatic territories [7]. Therefore, the draining LC was considered "predictable" for 16 selected anatomical regions (6 from the head and 10 from the hind limb) and "unpredictable" for 4 (all from the hind limb) (Tables 1 and 2).

## NIRF-ICG lymphography

After the drawing of one selected cutaneous region, the NIRF-ICG lymphography procedure included intradermal injection of ICG, visualization of the draining lymphatic pathways, and identification and surgical exploration of the draining LC(s) under NIRF camera guidance (Fig 3). In each cadaver, NIRF-ICG lymphography was performed in a maximum of three selected cutaneous regions, one from each anatomical district, i.e., one from the right hind limb, one from the left

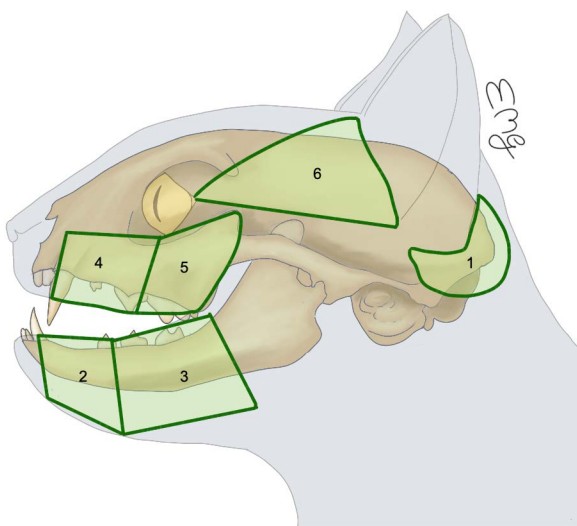

**Fig 1. Predetermined cutaneous regions of head anatomical district.** 1-Auricular; 2-Rostral-mandibular; 3- Aboral mandibular; 4- Rostral maxillary; 5- Aboral maxillary; 6- Temporal-zygomatic.

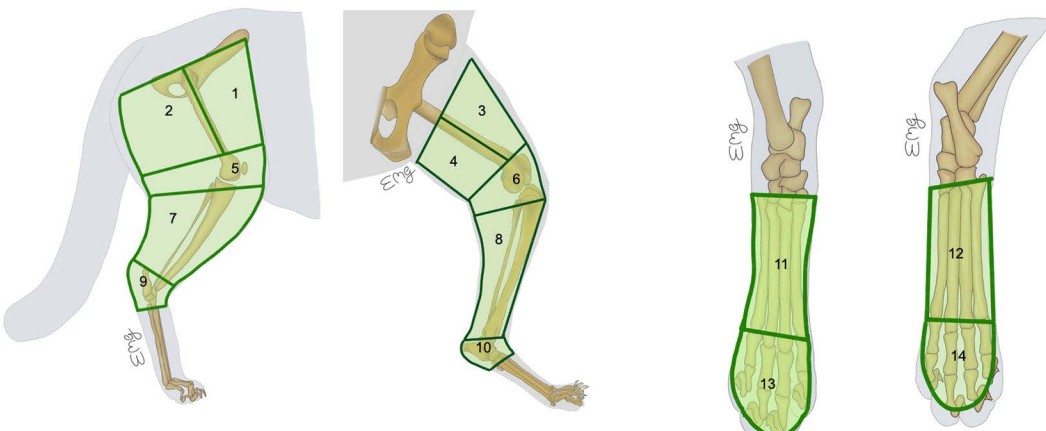

**Fig 2. Predetermined cutaneous regions of hind limb anatomical district.** 1- Lateral thigh – cranial; 2- Lateral thigh – caudal; 3- Medial thigh – cranial; 4- Medial thigh – caudal; 5- Lateral genicular; 6- Medial genicular; 7- Lateral crural; 8- Medial crural; 9- Lateral tarsal; 10- Medial tarsal; 11- Dorsal metatarsal; 12- Plantar metatarsal; 13- Dorsal phalangeal; 14- Plantar phalangeal.

hind limb, and one from the head. The side of each selected cutaneous region was randomly assigned using an online randomization tool (https://www.random.org/lists/). Considering the two hind limb regions selected per cadaver, to prevent lymphatic drainage from crossing the ventral midline, NIRF-ICG lymphography was performed firstly on the most distal region (either on the left or right side) and then on the opposite leg in the more proximal one.

For each lymphography, each cadaver was placed in lateral recumbency, and intradermal injections in multiple spots of indocyanine green at a distance of 0.5 cm from each other were performed to cover the whole selected cutaneous region. A total volume of 0.4 ml of ICG at 1,25 mg/ml (0.2 ml of ICG at 2.5 mg/ml diluted in 0.2 ml NaCl 0.9%) was used [7,10,29–31], and injection of full volume or volume reduction was recorded. At each injection spot, the needle was pointed towards the center, avoiding going beyond the margins of the selected cutaneous region. At the moment of the injection, room lights

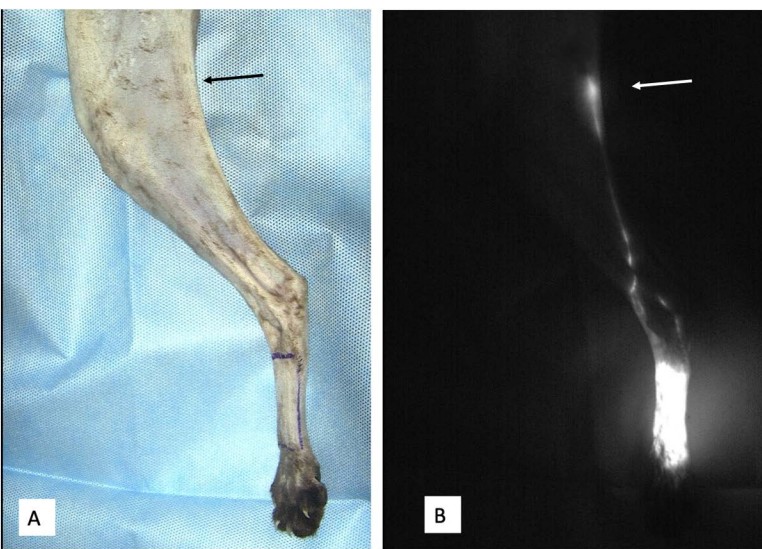

**Fig 3. NIRF-ICG lymphography images of the hind limb. A)** The dorsal metatarsal selected cutaneous region is drawn with a skin marker before performing NIRF-ICG lymphography. The anatomical location of the popliteal LC is pointed out by the black arrow; **B)** NIRF-ICG lymphography images after injection of ICG and massage of the selected cutaneous region showing the lymphatic drainage reaching the popliteal LC (white arrow).

were turned off, a timer was started from the last injection of dye, and a NIRF-camera (SPY-Elite Imaging system, Stryker, MIDA Tecnologia medica S.p.A) was positioned 20–30 cm above selected cutaneous region, recording the NIRF-ICG lymphography. Considering that *in vivo* lymphatic circulation works differently than in cadavers, being a dynamic process assisted by anatomical structures and valves, in the present study it was favored by gentle massage and, for the hind limbs, by flexion-extension movements of the limb for 10 minutes. To avoid contaminating the skin around the injected region with ICG, a gauze was used to absorb any residue or reflux from the injection sites, and operators changed their gloves after each injection and any time contamination with ICG was suspected. The LCs detected by NIRF-ICG lymphography were dissected and explored under the guidance of a NIRF-camera. In the case of intra-abdominal LC, when the detected lymphatic pathway was supposed to run through the inguinal canal, a ventral midline incision was performed, and the abdominal cavity was always surgically explored to detect any intra-abdominal fluorescence.

If no drainage or fluorescent LC was visible after the first 10 minutes, the injections at the same dosage and massage procedures were repeated. Ten minutes after the second injection, if the beginning of drainage was detected, massage of the region was continued for another 10 minutes to a total maximum time of 30 minutes. If that drainage did not progress or no fluorescent drainages or LCs were detected, expected regional LCs were surgically inspected under the guidance of NIRF-camera, and the LNs were removed.

## Lymph nodes evaluation

- All extirpated LNs were assessed by NIRF-camera. LNs were defined as "fluorescent" if they were either completely fluorescent or partially fluorescent, if fluorescence of the draining lymphatic vessel arrived at the LN and involved at least one pole of the node. Fluorescence was defined as signal-to-background ratio (SBR) > 1.1 [32].

- Extirpated LNs were measured by caliper on two orthogonal axes (millimeters).

- Further data recorded included: median number of LNs per LC, presence of a single or multiple LNs per LC overall and within the anatomical district.

Whenever a fluorescent LN was found, the corresponding LC was designated as draining the assessed selected cutaneous region.

## NIRF-ICG lymphography evaluation

- Success: the success of NIRF-ICG lymphography was defined as the detection, after surgical exploration guided by the NIRF-camera, of at least one fluorescent LN in a LC with or without transcutaneous visualization of a fluorescent lymphatic pathway.

- Failure: the NIRF-ICG lymphography was recorded as failed if no fluorescence could be visualized in any LN or LCs, or if only an interrupted lymphatic pathway was detected after a maximum of total 30 minutes of massage.

- Interrupted lymphatic pathways: their definition was used by the authors with a pure descriptive intent and was based only on visible macroscopical evidence, they appeared as fluorescent pathways departing from the injected selected cutaneous region and not arriving at any LC. To confirm the interruption of visible lymphatic pathways, the sites where drainages stopped and the expected and/or closest regional LCs were always surgically explored under NIRF-camera guidance to confirm the absence of fluorescence.

- Shine-through effect: if the fluorescence of the selected cutaneous region overlapped the fluorescent signal from the LC due to their anatomical proximity, it was defined and recorded as the "shine-through effect" [33]. In that case the LC fluorescence was assessed by post excision fluorescence of at least one the LN belonging to the LC.

## Variables collected related to NIRF-ICG lymphography

Variables collected and measured were: 1) the presence or absence of cutaneous pigmentation; 2) volume of ICG injected (full vs half); 3) the number of multiple injection spots needed to fill a selected cutaneous region; 4) the need to perform re-injections; 5) presence or absence of a drainage pathway visible transcutaneously; 6) the lymphatic pathway length (in centimeters), estimated by measuring with a tape the distance between the selected cutaneous regions and the identified LCs; 7) migration time (in minutes).

Data about migration time and lymphatic pathways length in case of "shine-through effect" were not collected. For each NIRF-ICG lymphography, the number, anatomical location, and side (ipsilateral *vs.* contralateral) of identified LC were recorded.

Considering the absence so far of a specific study on cats, correspondence with the predicted LC based on canine study [7] was assessed and defined as: total correspondence, if all predicted LCs were identified by fluorescence; partial correspondence, if fluorescence was identified, in addition to the expected one, in at least one LC at an unpredictable site; non-correspondence, if all LCs identified by fluorescence differed from the predicted ones.

## Statistical analysis

All data recorded were imported in a dedicated statistical analysis software (JMP®, v.16.0, SAS Institute, Cary, NC, USA; Prism 8 for MacOS, v. 8.2.0, GraphPad Software Inc., CA, USA). Categorical variables were reported as percentage (%) of the whole sample. Continuous variables were tested for normality with the Shapiro Wilk's $W$ test and reported as mean ± standard deviation or median and range, accordingly.

Association between categorical variables was assessed with contingency tables and Fisher's exact test or chi-squared test, while a univariate logistic regression model assessed the association between continuous and categorical variables.

In particular, the association between the following variables were assessed:

- success or failure of NIRF-ICG lymphography and:

  ◦ BCS class

  ◦ storage condition (refrigerated *vs.* frozen)

  ◦ anatomical district (head *vs.* hind limb)

  ◦ selected cutaneous regions

  ◦ presence of cutaneous pigmentation

  ◦ volume of ICG injected (full *vs.* half)

  ◦ number of multiple injection spots

- anatomical districts and technical aspects and results of lymphography, i.e.,:

  ◦ need to re-inject ICG

  ◦ volume of ICG injected (full *vs.* half)

  ◦ lymphatic pathway length

  ◦ migration time

The same associations were explored for the selected cutaneous region within the anatomical districts.

- presence of interrupted lymphatic pathways and:

  ◦ anatomical district (head *vs.* hind limb)

  ◦ selected cutaneous regions within the anatomical districts

  ◦ storage conditions (refrigerated *vs.* frozen)

- Storage condition (refrigerated *vs.* frozen) and:

  ◦ Migration time

- Need to re-inject and:

  ◦ volume injected

- Correspondences with lymphosomes in dog [7] and:

  ◦ anatomical district (head *vs.* hind limb)

  ◦ selected cutaneous regions within anatomical districts.

All significant association were further analyzed by Mann-Whitney U test between the anatomical districts or by Kruskal-Wallis test between the selected cutaneous regions. Each region was compared with all the others *post hoc* with Tukey's HSD test. For all tests, significance was set at $P < 0.05$.

## Results

The same two operators performed all procedures in a dedicated room (EMG, AU). A total of 33 feline cadavers were included; 27 (82%) cadavers were frozen, and the remaining 6 (18%) were refrigerated. Out of 99 anatomical districts

(3 per cat), 15 (7 heads and 8 hind limbs) were excluded due to macroscopic alterations. A total of 84 NIRF-ICG lymphographies were performed within the remaining 84 anatomical districts, of which 26 heads and 58 hind limbs. Considering the 20 predetermined cutaneous regions, lymphography was performed 5 times in 4 of them (auricular, temporal-zygomatic, medial crural, plantar phalangeal), and 4 times in the remaining 16.

Details regarding overlaying pigmented skin, volume injected, number of multiple injection spots per selected cutaneous region, need to perform re-injection, median lymphatic pathways length and median migration time are summarized in Tables 3 and 4. In particular, half volume was injected in the rostral mandibular region four times, in the aboral maxillary and dorsal phalangeal regions two times, and one each in the aboral mandibular, rostral maxillary, lateral tarsal, medial tarsal, and plantar metatarsal regions.

The NIRF-ICG lymphography was successful in 63/84 (75%) selected cutaneous regions. Conversely, the NIRF-ICG lymphography failed in the remaining 21/84 (25%) cases related to 11 selected cutaneous regions. Successful NIRF-ICG lymphography was always recorded in the following 9 selected cutaneous regions: rostral and aboral maxillary, rostral and aboral mandibular, dorsal phalangeal, dorsal and plantar metatarsal, lateral tarsal and medial thigh-cranial (Tables 5 and 6). Further details regarding NIRF-ICG lymphography for each selected cutaneous region and anatomical district are reported in Tables 4–6. An interrupted lymphatic pathway was observed in 13/84 (16%) NIRF-ICG lymphographies. Details on the interrupted lymphatic pathway are reported in S1 Table and Table 4.

Data about detected LCs, correspondence with predicted LCs, shine-through effect and detected LNs are reported in Tables 5–6; S2 Table.

A total of 114 LNs within the draining LCs were excised; the median number of LN per LC was 1 (1 − 5). Median LNs size was 5 (1–22) mm and 5 (1–17) mm considering major and minor axis, respectively. The post-excisional NIRF-camera assessment identified 96 (84%) fluorescent LNs and 18 (16%) non-fluorescent LNs.

## Association between success or failure of NIRF-ICG lymphography and variables collected

The variables evaluated for association with the success and the failure of NIRF-ICG lymphography are summarized in S3 Table. In particular, the failure of NIRF-ICG lymphography was associated with the selected cutaneous regions (P = 0.02), with the auricular and medial crural regions displaying the highest failure rate.On the other hand, lymphographies performed with half volume never displayed failures (P = 0.02).

## Association between anatomical districts/selected cutaneous regions and technical aspects and results of lymphography

Data on the associations between technical aspects, performances, and results of the NIRF-ICG lymphography – both among themselves and with anatomical districts – are summarized in Table 3 as well as the same associations comparing the selected cutaneous regions within the anatomical districts.

Specifically, the number of lymphographies that needed re-injection did not differ between the head and the hind limb (P = 0.09). Considering selected cutaneous regions within the head, the temporal-zygomatic region needed re-injection in 100% of lymphographies, whereas rostral maxillary and rostral and aboral mandibular regions never needed re-injection (P = 0.002). On the other hand, no difference in lymphographies that needed re-injection could be detected between the selected cutaneous regions belonging to the hindlimb (P = 0.61). When the possible association between the need to re-inject and the volume injected was evaluated, a significantly lower (P = 0.04) number of lymphographies needed re-injection when injecting half volume.

Considering the volume of ICG injected, the use of half the volume of ICG was significantly more frequent in the head compared to the hind limb (P = 0.009). Indeed, the rostral mandibular region was injected with half volume in 100% of lymphographies (P = 0.01). Conversely, no difference in the use of half or full ICG volume could be detected among the selected cutaneous regions belonging to the hindlimb (P = 0.21). Nonetheless, although non-significant, it should be noted

**Table 3. Description of median number of multiple injection spots, need to perform re-injection, median lymphatic pathways length, median migration time and volume of ICG injected for each selected cutaneous region.**

| Selected Cutaneous Region (n=84) | Median n. of multiple injection spots Overall, 8 (3–20) | Need to perform re-injection (n=48) | Lymphatic pathways length (cm) Overall, 4 (0–30) | Median migration time (min) Overall, 8 (1–30) | volume of ICG injected (full *vs.* half) |
|---|---|---|---|---|---|
| **Auricular** (n=5) | 8 (6 –10 )[A,B,C,D] | 4/5 | 2 (2 –2 )[A,B] | 2 (2 –2 )[A,B] | 5/5 full |
| **Rostral mandibular** (n=4) | 5 (4 –8 )[A] | 0/4* | 4 (2 –6 )[A,B] | 2 (1 –4 )[A] | 4/4 half* |
| **Aboral mandibular** (n=4) | 5 (4 –12 )[A,B] | 0/4* | 1 (0–2)[A] | 3 (2 –10 )[A,B] | 3/4 full 1/4 half |
| **Rostral maxillary** (n=4) | 6 (3 –12 )[A,B] | 0/4* | 4.5 (4 –5 )[A,B,C] | 4 (1 –10 )[A,B,C] | 3/4 full 1/4 half |
| **Aboral maxillary** (n=4) | 5 (4 –9 )[A] | 2/4 | 4 (4 –14 )[B,C] | 15 (8 –20 )[D,E] | 2/4 half 2/4 full |
| **Temporal-zygomatic** (n=5) | 8 (6 –8)[A,B] | 5/5 | 3 (1 –6 )[A,B] | 16 (8 –20 )[D,E] | 5/5 full |
| **Lateral thigh – cranial** (n=4) | 14 (6 –16 )[E] | 3/4 | 2 (1 –2 )[A,B] | 15 (10 –20 )[D,E] | 4/4 full |
| **Lateral thigh – caudal** (n=4) | 8 (6 –16)[A,B,C,D,E] | 3/4 | 1 (1–1)[A,B] | 20 (20–20)[D,E] | 4/4 full |
| **Medial thigh – cranial** (n=4) | 10 (8 –14 )[A,B,C,D,E] | 2/4 | 1 (0–3)[A] | 10 (7–25)[D,E] | 4/4 full |
| **Medial thigh – caudal** (n=4) | 11 (7 –14 )[B,C,D,E] | 3/4 | 1.5 (1 –5 )[A,B] | 27 (5–30)[E] | 4/4 full |
| **Lateral genicular** (n=4) | 11 (8 –17 )[C,D,E] | 4/4 | 1 (0–7)[A,B] | 9 (5 –11 )[A,B,C,D,E] | 4/4 full |
| **Medial genicular** (n=4) | 10 (8 –20 )[D,E] | 3/4 | 0.5 (0–1)[A,B] | Shining effect | 4/4 full |
| **Lateral crural** (n=4) | 12 (12 –15 )[E] | 1/4 | 2 (2 –2 )[A,B] | 2 (2 –8 )[A,B,C,D] | 4/4 full |
| **Medial crural** (n=5) | 12 (7 –15 )[C,D,E] | 4/5 | 2 (2 –2 )[A,B] | 10 (10 –25 )[A,B,C,D,E] | 5/5 full |
| **Lateral tarsal** (n=4) | 7 (6 –10 )[A,B,C,D] | 2/4 | 10 (7–25)[D,E] | 11 (5 –20 )[B,C,D] | 3/4 full 1/4 half |
| **Medial tarsal** (n=4) | 7 (4 –8 )[A,B] | 3/4 | 7 (7 –15 )[C,D] | 7 (7 –17 )[A,B,C,D] | 3/4 full 1/4 half |
| **Dorsal metatarsal** (n=4) | 7 (6 –12 )[A,B,C,D] | 2/4 | 10 (9 –10 )[C,D] | 7 (1 –15 )[A,B,C,D] | 4/4 full |
| **Plantar metatarsal** (n=4) | 9 (6 –12 )[A,B,C,D,E] | 1/4 | 14 (11 –25 )[E,F] | 8 (2–20)[A,B,C,D] | 3/4 full 1/4 half |
| **Dorsal phalangeal** (n=4) | 8 (3 –10 )[A,B] | 2/4 | 15 (11–30)[F] | 14 (2–30)[C,D,E] | 2/4 full 2/4 half |
| **Plantar phalangeal** (n=5) | 8 (6 –10 )[A,B,C] | 4/5 | 14 (11 –24 )[E,F] | 8 (2–20)[A,B,C,D] | 5/5 full |

Legend: LC=lymphocentrum; Statistics: within columns, measures not connected by the same letter are significantly different at P<0.05. The *indicates significant difference at chi-square test.

that only the lateral and medial tarsal regions (one lymphography each), dorsal phalangeal (two lymphographies) and plantar metatarsal (one lymphography) needed half-volume injection.

The lymphatic pathway length differed significantly between the head and the hind limb (P=0.0006). Indeed, the lymphatic pathway was significantly shorter (P=0.0002) in the head (4; 1–20 cm) compared to the hind limb (10; 1–30 cm). The selected cutaneous regions belonging to the head differed significantly from each other (P=0.006), with the aboral maxillary region displaying a longer lymphatic pathway compared to the other regions. Indeed, in one lymphography of the aboral maxillary region, the identified LC was the superficial cervical node. The selected cutaneous regions belonging to the hind limb differed significantly from each other (P=0.003), with the dorsal and plantar phalangeal and plantar metatarsal regions displaying the longer lymphatic pathway compared to the other regions. In one lymphography of the dorsal and two lymphographies of the plantar phalangeal region, one of the identified LC was the medial iliac. Conversely, in one lymphography of the plantar metatarsal region, one of the identified LC was the superficial inguinal.

As for the migration time, it differed significantly between the head and the hind limb (P=0.024). In particular, migration time was significantly shorter (P=0.026) in the head (4; 1–20 min) compared to the hind limb (10; 1–30 min). The selected cutaneous

**Table 4. Details on lymphographies divided by anatomical districts.**

| NIRF-ICG Lymphography | Head (26 lymphographies) | Hind Limb (58 lymphographies) | Overall (head + hind limbs) (84 lymphographies) |
|---|---|---|---|
| **Success** | 21/26 (81%) | 42/58 (72%) | 63/84 (75%) |
| **Failure** | 5/26 (19%) | 16/58 (28%) | 21/84 (25%) |
| **Interrupted Lymphatic Pathways**[*] | 2/26 (8%) | 11/58 (19%) | 13/84 (16%) |
| **Overlaying pigmented skin** | 6/26 (23%) | 4/58 (7%) | 10/84 (12%) |
| **Volume injected (full *vs*. half)** | 18/26 (69%) full | 53/58 (91%) full | 71/84 (85%) full |
| | 8/26 (31%) half | 5/58 (9%) half | 13/84 (15%) half |
| **Median n. multiple injection spots** | 7 (3 –12 ) | 8 (3-20) | 8 (3-20) |
| **Need to perform re-injection** | 11/26 (42%) | 37/58 (64%) | 48/84 (57%) |
| **Median lymphatic pathways length** (cm) | 4 (1 –14 ) | 10 (1-30) | 5 (1-30) |
| **Median migration time** (min) | 4 (1-20) | 10 (1-30) | 8 (1-30) |

[*]In 9/13 (69%) lymphographies, despite the interrupted drainage, another lymphatic pathway properly reached the draining LC. In the remaining 4/13 (31%), the interrupted lymphatic pathway was the only drainage detected, therefore these lymphographies were counted in the 21 that failed. In the head anatomical district, despite the interrupted drainage, another lymphatic pathway properly reached the draining LC.

**Table 5. Correspondence between selected cutaneous regions with expected draining LCs in dogs [7]– HEAD.**

| Selected Cutaneous Region | Success of Lymphography | Detected LCs | Expected LCs in dogs | Correspondence with dogs |
|---|---|---|---|---|
| **Auricular** (n = 5) | Successful: 1/5 | 1/1 Mandibular + Parotid + Superficial Cervical | Superficial cervical | Partial |
| **Rostral mandibular** (n = 4) | Successful: 4/4 | 2/4 Mandibular | Mandibular | Total |
| | | 2/4 Mandibular + Medial retropharyngeal | | Partial |
| **Aboral mandibular** (n = 4) | Successful: 4/4 | 3/4 Mandibular | Mandibular | Total |
| | | 1/4 Mandibular + Parotid | | Partial |
| **Rostral maxillary** (n = 4) | Successful 4/4 | 4/4 Mandibular | Mandibular | Total |
| **Aboral maxillary** (n = 4) | Successful: 4/4 | 1/4 Mandibular | Mandibular | Total |
| | | 1/4 Mandibular + Parotid | | Partial |
| | | 1/4 Mandibular + Parotid + Medial Retropharyngeal | | Partial |
| | | 1/4 Superficial cervical | | Non-correspondent |
| **Temporal-Zygomatic** (n = 5) | Successful: 4/5 | 3/5 parotid | Parotid | Total |
| | | 1/5 parotid + mandibular | | Partial |

Legend: LC = Lymphocentrum; correspondence was defined as: TOTAL = all LC hypothesized based on dogs were identified by fluorescence, PARTIAL = fluorescence identified at least one more LC at an unpredictable site; NON-CORRESPONDENT = all LCs identified by fluorescence were different than the hypothesized ones.

regions belonging to the head differed significantly from each other (P = 0.001), with a longer migration time in the aboral maxillary and the temporal-zygomatic regions. Particularly, in the aforementioned lymphography of the aboral maxillary region, migration time might have been affected by the longer lymphatic pathway length. The selected cutaneous regions belonging to the hind limb did not differ between each other for migration time (P = 0.69). Finally, considering storage conditions, migration time was significantly lower (P = 0.04) in refrigerated (2; 1–20 min) cats compared to frozen (10; 1–30 min) ones.

## Association between presence of interrupted lymphatic pathways and variables collected

The identification of an interrupted lymphatic pathway was not associated with the anatomical district (P = 0.33), nor to the selected cutaneous regions whether belonging to the head (P = 0.43) or to the hind limb (P = 0.68). It was also not associated to the storage condition (P = 0.80).

**Table 6. Correspondence between selected cutaneous regions with expected draining LCs in dogs [7]– HIND LIMB.**

| Selected Cutaneous Region | Success of Lymphography | Detected LCs | Expected LCs in dogs | Correspondence with dogs |
|---|---|---|---|---|
| **Lateral thigh – cranial** (n=4) | Successful: 3/4 | 2/4 Superficial inguinal | Superficial inguinal | Partial |
| | | 1/4 Popliteal | Medial Iliac | Non-correspondent |
| **Lateral thigh – caudal** (n=4) | Successful: 2/4 | 2/4 Popliteal | Superficial inguinal | Non-correspondent |
| | | | Medial Iliac | |
| | | | Sacral | |
| **Medial thigh – cranial** (n=4) | Successful: 4/4 | 4/4 Superficial inguinal | Superficial inguinal | Total |
| **Medial thigh – caudal** (n=4) | Successful: 3/4 | 1/4 Popliteal + Superficial inguinal | Superficial inguinal | Partial |
| | | 1/4 Popliteal | | Non-correspondent |
| | | 1/4 Popliteal + Medial iliac | | Non-correspondent |
| **Lateral Genicular** (n=4) | Successful: 3/4 | 1/4 Popliteal + Superficial inguinal | Superficial inguinal | Partial |
| | | 1/4 Popliteal | | Non-correspondent |
| | | 1/4 Popliteal + Medial iliac | | Non-correspondent |
| **Medial Genicular** (n=4) | Successful: 2/4 | 2/4 Popliteal | Superficial inguinal | Non-correspondent |
| **Lateral Crural** (n=4) | Successful: 3/4 | 3/4 Popliteal | Popliteal | Partial |
| | | | Superficial inguinal | |
| **Medial Crural** (n=5) | Successful: 1/5 | 1/5 Popliteal + Superficial inguinal | Superficial inguinal | Partial |
| **Lateral Tarsal** (n=4) | Successful: 4/4 | 3/4 Popliteal | Popliteal | Partial |
| | | 1/4 Popliteal + Medial iliac | Superficial inguinal | Partial |
| **Medial Tarsal** (n=4) | Successful: 2/4 | 1/4 Popliteal + Medial iliac | Superficial inguinal | Non-correspondent |
| | | 1/4 Medial iliac | | Non-correspondent |
| **Dorsal Metatarsal** (n=4) | Successful: 4/4 | 4/4 Popliteal | Popliteal | Total |
| **Plantar Metatarsal** (n=4) | Successful: 4/4 | 3/4 Popliteal | Popliteal | Total |
| | | 1/4 Popliteal + Superficial inguinal | | Partial |
| **Dorsal Phalangeal** (n=4) | Successful: 4/4 | 3/4 Popliteal | Popliteal | Total |
| | | 1/4 Popliteal + Medial iliac | | Partial |
| **Plantar Phalangeal** (n=5) | Successful: 3/5 | 1/5 Popliteal | Popliteal | Total |
| | | 1/5 Popliteal + Medial iliac | | Partial |
| | | 1/5 Medial iliac | | Non-correspondent |

Legend: LC=Lymphocentrum; correspondence was defined as: TOTAL=all LC hypothesized based on dogs were identified by fluorescence, PARTIAL=fluorescence identified at least one more LC at an unpredictable site; NON-CORRESPONDENT=all LCs identified by fluorescence were different than the hypothesized ones.

## Association between correspondence with lymphosomes in dogs [7] and variables collected

When correspondence with lymphosomes in dogs [7] was explored, it differed significantly between the anatomical districts (P=0.049), with the head displaying a lower number of non-correspondences. No differences could be detected in correspondence with lymphosomes in dogs [7] between the selected cutaneous regions belonging to the head (P=0.40), while within the hind limb, the medial tarsal, medial genicular, and lateral thigh-caudal regions displayed only non-correspondence, whereas the lateral tarsal, lateral and medial crural regions showed only partial correspondence (P=0.0004).

## Discussion

The results of this exploratory study demonstrated the feasibility of NIRF-ICG lymphography for superficial lymphatic pathways definition in the head and hind limb anatomical districts of feline cadavers. We obtained a 75% overall success rate (81% for the head and 72% for the hind limb). Beyond feasibility, this study differs from previous ones due to the higher

number of repetitions, the larger sample size, and the broader range of variables explored, allowing for consideration of technical aspects that may influence lymphography performance [2–7]. While lymphography in cats has been reported to be successful with other tracers, such as isosulfan blue or India ink [5,6,19], to the authors' knowledge, only Ratzlaff's study investigated the superficial lymphatic system, but in living cats, euthanizing them after injection of dye. Although there are limitations compared to studies on live animals, the approach reported in the present paper offered interesting results without compromising animal welfare and respecting the ethical principles of research.

The limited number of studies on the superficial lymphatic system of cats makes it challenging to compare our results with previously published data, given the differing inclusion criteria and the numerous variables explored in our study. Considering our results, no association was found between the success of NIRF-ICG lymphography and the variables considered, neither related to the cadaver itself (BCS, storage conditions, anatomical districts, selected cutaneous regions, cutaneous pigmentation) nor the technique applied (number of multiple injection spots, need to perform re-injection, injected volume). Unfortunately, these variables were not explored in previous studies, thus preventing comparison. Similarly, no information about failure has been reported in literature so far [2–7,19], but in the present study failure of NIRF-ICG lymphography resulted significantly associated to selected cutaneous regions. In particular, the auricular and medial crural regions showed the highest failure rate, with 4/5 failures. On one hand, the high failure rate recorded for the auricular region could be due to the difficulty in massaging the regions and the inability to perform any flexion-extension movement. Additionally, the auricular region may have less elastic skin and underlying tissues than other anatomical areas, with a lower ability to deform or move freely. This could prevent the correct massaging and uniform diffusion of the dye. On the other hand, no factors have been identified for the medial crural region that would support a scientifically sound hypothesis. Difficulties in massaging and moving the area further may contribute to tracer failure, but further studies on a higher number of samples may help to find an explanation to this result.

Moreover, the failure rate of NIRF-ICG lymphography was lower when half volume of dye was injected, although the concentration of ICG injected remained the same. A low number of half-volume injections were performed, possibly biasing statistical analyses. Still, this result may suggest that a cutaneous NIRF-ICG lymphography in feline cadavers would be successful regardless of the volume injected, at least for the volumes and ICG concentration used in this study.

Besides the success and failure rates, the performance characteristics of the NIRF-ICG lymphography and variables potentially influencing them were deeply explored in the present study. Among them, the storage conditions were associated with migration time, which was shorter in refrigerated cadavers compared to frozen ones, even if this result should be contextualized in light of a small number of refrigerated cadavers included in this study. However, the migration time recorded in feline cadavers, either refrigerated or frozen, was higher than that reported in studies on live healthy cats and dogs [34,35]. This could be explained by the hypothesis that microscopic cadaveric alterations have influenced the integrity of superficial lymphatic vessels, slowing down ICG migration. Indeed, in canine cadavers, storage conditions have been reported to affect the integrity of lymphatic vessels [36]. Furthermore, migration time and the lymphatic pathways length, were associated with the anatomical districts and selected cutaneous regions. The fact that migration time was shorter in the head rather than in the hind limb might be explained by anatomical characteristics: in fact, selected cutaneous regions of the hind limb are often more distant from their detected LC compared to the regions of the head. Anyway, the difference in migration time between selected cutaneous regions of the head may also depend on difficulty in massaging and migration due to reduced presence of underlying soft tissues (temporal-zygomatic regions) and to the migration from one aboral mandibular region to a superficial cervical LC that may have influenced the migration time. Considering the lymphatic pathways length recorded, the most interesting finding concerned some regions of the distal portion of hind limb. As discussed later, a selected cutaneous region might physiologically drain toward one or more LCs, even at a long distance. Hence, the distance recorded in the present study varied highly even within each selected cutaneous region, most likely due to individual differences. In particular, the plantar phalangeal region in two lymphographies, and the dorsal phalangeal, lateral, and medial tarsal regions in one lymphography each displayed ICG migration to the medial iliac LC

(alone or in association with the popliteal LC) always bypassing the superficial inguinal LC, determining a significantly longer lymphatic pathway.

Concerning technique characteristics, more than half (57%) of NIRF-ICG lymphographies needed re-injections, suggesting that NIRF-ICG lymphography on feline cadavers may be less immediate than *in vivo* [34]. In particular, the temporal-zygomatic region needed re-injection in 100% of its NIRF-ICG lymphographies, hampering, together with the aforementioned longer migration time, the hypothesis of a more difficult migration of the tracer for this specific cutaneous region. The need to re-inject has been previously reported in a canine cadaveric study, with up to four re-injections [10]. Overall, ICG migration may need more time and even re-injection of the dye to happen in cadavers. In some instances, a reduction of the injected volume was needed. Half volume was used when the predetermined region was completely filled before the total volume was injected. Since some small regions received the full volume while others received half, clinical and anatomical factors such as the level of dehydration, the relative thickness of the cutis and subcutis, and the tight adherence to underlying tissues (as seen in the head region, where most half-volume injections were recorded) may have influenced the need to reduce the volume. In fact, while certain small regions tolerated the full volume, others required only half.

The association of half volume and a lower number of re-injections identified in the present study confirms the aforementioned hypothesis that the amount of ICG injected might not be determining the success of NIRF-ICG lymphographies.

The phenomenon of interrupted lymphatic pathways has not been previously described in lymphographic studies [2,7] or studies on the feasibility of NIRF-ICG lymphography in human cadavers [8,37,38]. In the present study, interrupted lymphatic pathways were mostly detected in the hind limb district (84%), but this was also the anatomical district including a higher number of selected cutaneous regions and lymphographies. Anyway, interrupted lymphatic pathways were not associated to any of the variables explored, and it could be supposed that the interruption of migration may be linked to peculiar anatomical characteristics or post-mortem alterations.

In the present study, in 56% of overall successful NIRF-ICG lymphographies, the selected cutaneous regions displayed a detected LC partially or totally non-correspondent, thus confirming the prediction of the draining LC in cats based on canine studies could be misleading. Correspondence was associated with the anatomical district, with the head showing the highest number of correspondences, with mandibular LC representing the more frequently detected, and the only one for the rostral maxillary region. This result contrasts with what is reported in the literature regarding the complexity of the superficial lymphatic system in this anatomical district [12,39,40]. However, the decision in this study to exclude the median regions may have significantly influenced the outcome, and thus, this result should be interpreted with caution.

Among the interesting findings regarding the identified LCs, the auricular, aboral mandibular, aboral maxillary regions migrated to the parotid LC. This is in partial contrast with what has been reported in cadaveric studies of dogs and cats, in which the parotid LC was considered to drain only the temporal-zygomatic, ocular and auricular area [7,5,41]. In the present study, three NIRF-ICG lymphographies (2 rostral mandibular, and 1 aboral maxillary) showed migration to medial retropharyngeal LC in addition to the expected one. Since it is considered the collector of all LCs of the head [41,42] and it was never the sole draining LC in the present study, the medial retropharyngeal LC might be considered as a 2nd-tier LC [43]. Lastly, one aboral maxillary NIRF-ICG lymphography displayed migration only to the superficial cervical LC. This migration seemed quite peculiar and has not been previously reported either in cats or in dogs [5,7].

Concerning the hind limb, in the present study, total correspondence was reported in 36% of successful NIRF-ICG lymphographies with only the medial thigh-cranial and dorsal metatarsal regions displaying 100% migrations to the same expected LC (superficial inguinal and popliteal, respectively). Six NIRF-ICG lymphographies from different selected cutaneous regions, also belonging to the lower hind limb, distal to the knee, showed the medial iliac as unexpected draining LC. This is in contrast to what is reported in the dog [7], with the medial iliac LC draining mainly the cranial part of the

lateral thigh, whereas it is in partial agreement with Ratzlaff [5], who reported the presence of a drainage pathway running from the lateral distal hind limb, bypassing the popliteal LC and directly getting to the iliac LCs. A possible explanation might be the lack of separation between the superficial and deep lymphatic system in the distal part of the hind limb, as reported in the rabbit [3]. In this case, the dye might have reached the deep lymphatic system, leading to different lymphatic pathways.

In the present study, medial and lateral genicular, medial thigh-caudal, lateral thigh-cranial and caudal regions displayed migration only to the popliteal LC, in contrast to what was reported in the dog [7]. This is intriguing, since these regions are in proximity or even proximally to the popliteal LC and for which the expected draining LCs were the superficial inguinal, medial iliac, or sacral. A similar finding was reported by Ratzlaff [5], but in that study, it was limited to the lateral compartment of the thigh.

The present study might be considered based on a clinical setting rather than a purely descriptive anatomical study. Anatomical landmarks were chosen to be easily assessable during clinical practice, mostly by palpation, while in previous studies [3–5,7] cutaneous regions were not univocally defined by landmarks, and they were deduced according to the draining LC. Furthermore, in the present study, NIRF-ICG lymphography was chosen since ICG can be injected intradermally, mimicking mapping procedures of sentinel LCs in vivo [22,29,35,44,45]. As mentioned before, both Ratzlaff [5] and Suami [7] injected multiple regions per lymphographic study, while in the present study, a single selected cutaneous region from the head and the two hind limbs was assessed in each cadaver. This allowed us to avoid the overlapping of lymphatic drainages coming from different selected cutaneous regions, leading to a univocal definition of LC drainage. Further differences included in the study design are related to the species involved, but also to the inclusion of a larger number of cadavers (33 cats in the present study vs 4 dogs in Suami [7]) and at least four repetitions of NIRF-ICG lymphographies for each selected cutaneous region. The univocal definition of the selected cutaneous regions by anatomical landmarks as well as increasing the number of NIRF-ICG lymphographies per selected cutaneous regions, were performed to improve repeatability and reliability and reduce the influence of individual cadaver variability.

Considering the limitations of this study, a relatively low number of refrigerated cadavers compared to the frozen ones has been studied, masking other possible associations. Even if this study included a higher number of lymphographies (84 NIRF-ICG lymphographies) than any previous study, only four to five lymphographies were repeated per region. A higher number of repetitions might confirm the identified draining patterns or even unveil new ones. Additionally, due to the high variability in the LCs detected by NIRF-ICG lymphography within the same selected cutaneous regions (such as the aboral maxillary, medial thigh-caudal, lateral genicular, and plantar phalangeal regions), there appears to be significant individual variation among cat cadavers included in this study. This variability prevented us from definitively describing predictable lymphatic draining patterns for all regions. The association of anatomical districts, and selected cutaneous regions within them, with several of the technical aspects investigated suggests that anatomical characteristics of the regions themselves may influence the NIRF-ICG lymphography results. Again, further studies with a larger number of repetitions for each selected cutaneous region may shed light on possible unidentified associations with these factors. In conclusion, based on the results of the present study, NIRF-ICG lymphography is feasible on selected cutaneous regions belonging to the head and hind limb of feline cadavers. These results suggest that superficial lymphatic drainage of the head and hind limb anatomical districts of cats may be complex to predict and seem different from what is reported in dogs. The use of repeatable anatomical landmarks can provide valuable assistance in determining lymphatic drainage from predetermined skin areas. However, the unpredictability of lymphatic drainage, combined with the high individual variability observed in this study, hampers the importance of using technologies such as NIRF-ICG for LC lymphography for the detection of SLCs in a clinical setting for staging and/or therapeutical purpose. Future studies might aim at mapping the whole superficial cutaneous lymphatic system, including unpaired median regions.

## Supporting information

**S1 Table.  Description of interrupted lymphatic pathways, the relative selected cutaneous region and detected draining LCs.** LC = Lymphocenter; *, ** = those interrupted lymphatic pathways were detected on the same cadaver, respectively.
(PDF)

**S2 Table.  Details about LC detected by lymphographies divided by anatomical district.** Only successful lymphographies were considered (21/26 head, 42/58 hind limb). LC = Lymphocentrum. In 1/63 NIRF-ICG lymphography, the detected LC (right mandibular) was contralateral to the selected cutaneous region side (left rostral mandibular).
(PDF)

**S3 Table.  Association between success and failure of NIRF-ICG lymphography and selected variables.**
BCS = Body Condition Score. Statistical association: P < 0.05.
(PDF)

## Author contributions

**Conceptualization:** Alessandra Ubiali, Elisa Maria Gariboldi, Damiano Stefanello.

**Data curation:** Alessandra Ubiali, Elisa Maria Gariboldi, Luigi Auletta, Damiano Stefanello.

**Formal analysis:** Luigi Auletta.

**Investigation:** Alessandra Ubiali, Elisa Maria Gariboldi, Damiano Stefanello.

**Methodology:** Alessandra Ubiali, Elisa Maria Gariboldi, Damiano Stefanello.

**Project administration:** Damiano Stefanello.

**Resources:** Alessia Di Giancamillo, Silvia Clotilde Bianca Modina.

**Supervision:** Damiano Stefanello.

**Writing – original draft:** Alessandra Ubiali, Elisa Maria Gariboldi, Luigi Auletta, Damiano Stefanello.

**Writing – review & editing:** Alessandra Ubiali, Elisa Maria Gariboldi, Luigi Auletta, Alessia Di Giancamillo, Silvia Clotilde Bianca Modina, Roberta Ferrari, Filippo Tagliasacchi, Valeria Martini, Damiano Stefanello.

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
