## [Decision Letter · Decision Letter 0]

Dear Dr. Ubiali,

Thank you for submitting your manuscript to PLOS ONE. After careful consideration, we feel that it has merit but does not fully meet PLOS ONE’s publication criteria as it currently stands. Therefore, we invite you to submit a revised version of the manuscript that addresses the points raised during the review process.

**Please address all Reviewer comments.**

We look forward to receiving your revised manuscript.

Kind regards,

Douglas H. Thamm, V.M.D.

Academic Editor

PLOS ONE

**Journal Requirements:**

1. When submitting your revision, we need you to address these additional requirements. Please ensure that your manuscript meets PLOS ONE's style requirements, including those for file naming. The PLOS ONE style templates can be found at https://journals.plos.org/plosone/s/file?id=wjVg/PLOSOne_formatting_sample_main_body.pdf and https://journals.plos.org/plosone/s/file?id=ba62/PLOSOne_formatting_sample_title_authors_affiliations.pdf

Reviewers' comments:

Reviewer's Responses to Questions

**Comments to the Author**

1. Is the manuscript technically sound, and do the data support the conclusions?

Reviewer #1: Yes

Reviewer #2: Partly

2. Has the statistical analysis been performed appropriately and rigorously?

Reviewer #1: Yes

Reviewer #2: I Don't Know

3. Have the authors made all data underlying the findings in their manuscript fully available?

Reviewer #1: Yes

Reviewer #2: Yes

4. Is the manuscript presented in an intelligible fashion and written in standard English?

Reviewer #1: Yes

Reviewer #2: No

**Reviewer #1:**  The authors should be commended for conducting a good study. The information provided in this manuscript, while based on cadavers, lays out the framework for the establishment of lymphatic drainage in cats. Future studies can focus on investigating the initial findings from this study and utilize that data in clinical settings. A few specific questions are below:

Abstract

-Lines 42-43: Since there are 20 cutaneous regions identified and you are using 33 cadavers, please include a comment about why only 84 regions were selected for evaluation. You explain this well in the paper, but this needs an explanation in the Abstract or you need to rephrase this section.

Discussion

-Page 26 Line 414: Could this lower failure rate be due to quenching of ICG that occurs when higher concentrations are administered? It would be worth considering this throughout all your comments in Lines 414-417.

**Reviewer #2:**  Overall an interesting study from a basic science perspective, but limited utility for clinical applications.

The authors' passion for the project is appreciated and acknowledged.

Why was the ICG diluted with saline for this project? How was the dose and volume determined prior to use in this study? This was not addressed in the Materials.

Table 7 should be omitted in my opinion. It seems the data is being manipulated too much to create conclusions (by only using the 'good' cases.)

Table 8 should also be omitted in my opinion or go into the supplement.

'innoculate' in line 455 doesn't make sense

omit whole paragraph on line 464.

the supplemental files do not add to the manuscript in my opinion.

A frustration I have with this study is the assumption that the lymphatic network is a series of empty open plumbing. There is no mention of how the normal lymphatic circulation really works, it is a dynamic process that requires energy for the flow of lymph, how particles enter the lymphatic circulation, nor how small and delicate the distal/terminal lymphatic vessels really are. In vivo when the distal lymphatic vessels become distended, this actually closes lymphatic valves.

I question the terminology of interrupted lymphatic pathways. Were these pathways really interrupted? Could it be reflux into another connected branching network of lymphatic vessels? Are there other possibilities--- please elaborate. Maybe there could be a statement for the value of histopathology to help figure it out?

It is commendable that the cat is the subject of this study as this species is understudied in the literature and is perhaps more common/numerous than dogs in real life. This study may influence clinical decision making for cancer surgery in cats, at least with the awareness of need for a larger anatomic geography when sentinel lymph node mapping is employed.

**Do you want your identity to be public for this peer review?** For information about this choice, including consent withdrawal, please see our Privacy Policy

Reviewer #1: No

Reviewer #2: No

---

## [Author Response · Author response to Decision Letter 1]

4 Jun 2025

Reviewer #1: The authors should be commended for conducting a good study. The information provided in this manuscript, while based on cadavers, lays out the framework for the establishment of lymphatic drainage in cats. Future studies can focus on investigating the initial findings from this study and utilize that data in clinical settings. A few specific questions are below:

We thank the R#1 for the positive comment, we think it goes straight towards the meaning of this paper and the research behind the text. We are glad that the scientific meaning of this work was well understandable and shared by the reviewer.

Abstract

-Lines 42-43: Since there are 20 cutaneous regions identified and you are using 33 cadavers, please include a comment about why only 84 regions were selected for evaluation. You explain this well in the paper, but this needs an explanation in the Abstract or you need to rephrase this section.

We agree with the reviewer, we added this information in the abstract text (lines 42-43): ICG-lymphography was repeated in 33 cadavers. “Out of the 99 selected cutaneous regions available, 15 were excluded following inclusion criteria, therefore lymphographies were performed for a total of 84 selected cutaneous regions (26 from the head and 58 from the hind limbs).”

Discussion

-Page 26 Line 414: Could this lower failure rate be due to quenching of ICG that occurs when higher concentrations are administered? It would be worth considering this throughout all your comments in Lines 414-417.

The authors believe that there is not enough numerosity to prove this hypothesis, which is, anyway, legitimate. Furthermore, we want to clarify that even when half of the volume was administered, the concentration remained the same; only the volume changed. We added this clarification to the main text (line 416).

Reviewer #2: Overall, an interesting study from a basic science perspective, but limited utility for clinical applications.

The authors' passion for the project is appreciated and acknowledged.

Thank you, we are glad that the reviewer wants to remark and acknowledge the aim of the paper. We undoubtedly agree that cadavers are not to be considered a concrete model for clinical studies, and is far from the aim of this paper to be considered part of it.

Why was the ICG diluted with saline for this project?

Thank you for the question.

The reported concentration and dilution of ICG with saline is the same method that we use in clinical practice, and was chosen in accordance with information retrieved in literature concerning ICG lymphography, both in vivo and in cadavers (Wan et al. 2021, Sanchez-Margallo et al. 2020).

How was the dose and volume determined prior to use in this study? This was not addressed in the Materials.

Considering that in literature a wide range is reported (from 0.5mg/ml to 2,5mg/ml; Favril et al. 2019, Townsend et al. 2017, Wan et al. 2021, Suami et al. 2013, Sanchez-Margallo et al. 2020, Arz et al. 2022; Beer et al. 2022), we defined a concentration of 1,25mg/ml, included in the aforementioned range.

We added the information about the concentration definition in the MM section, lines 157-158 “A total volume of 0.4ml of ICG at 1,25mg/ml (0.2 ml ICG at 2.5 mg/ml diluted in 0.2 ml NaCl 0.9%), (Sanchez-Margallo et al 2020; Wan et al. 2021; Suami et al 2013; Arz et al 2022; Beer et al 2022)”

Townsend KL, Milovancev M, Bracha S. Feasibility of near-infrared fluorescence imaging for sentinel lymph node evaluation of the oral cavity in healthy dogs. Am J Vet Res. 2018;79(9):995-1000. doi:10.2460/ajvr.79.9.995

Table 7 should be omitted in my opinion. It seems the data is being manipulated too much to create conclusions (by only using the 'good' cases.)

Thank you for the comment. We agree with the reviewer that table 7 does not give crucial information related to the aim of the paper. We want to clarify the aim of that table, since we are afraid its purpose might have been misunderstood. This table schematically describes the characteristics of the LC that were detected by lymphographies (i.e. number, whether they were multiple or not, how many LNs were detected within the same LC etc.). Failed lymphographies were forcibly excluded since the data were not available for those. Hence, we agree that data reported are not central for the manuscript, but we rather suggest not to omit the whole table, but to move it to the supplementary material, too.

Table 8 should also be omitted in my opinion or go into the supplement.

Thank you, we agree that since the main information regarding the association are reported in the text, this table can be moved into the supplementary material.

'innoculate' in line 455 doesn't make sense

Thank you for pointing it out, we changed it from “inoculated” to “injected”.

omit whole paragraph on line 464.

Done, thank you. We removed the sentence at lines 464-466.

the supplemental files do not add to the manuscript in my opinion.

We agree, we followed your suggestions and modified supplemental materials as follows:

- S1 Table was removed (signalment data) together with the sentence at lines 260-261 and 91-92.

- S3 Figure and S4 Figure were removed

- In the authors’ opinion, S2 Table should remain (now named S1 Table)

- Tables 7 and 8 were moved from the main text to Supplemental Material as table S2 and S3, respectively.

A frustration I have with this study is the assumption that the lymphatic network is a series of empty open plumbing. There is no mention of how the normal lymphatic circulation really works, it is a dynamic process that requires energy for the flow of lymph, how particles enter the lymphatic circulation, nor how small and delicate the distal/terminal lymphatic vessels really are. In vivo when the distal lymphatic vessels become distended, this actually closes lymphatic valves.

We understand the frustration of the Reviewer, and we agree with the considerations.

We added it in the text, lines 162-163 “Considering that in vivo lymphatic circulation works differently than in cadavers, being a dynamic process, assisted by anatomical structures as valves, in the present study it was favored by gentle massage and, for the hind limbs, by flexion-extension movements of the limb for 10 minutes.”

The choice to inject ICG intradermally was based on the fact that, as others authors did before (Suami et al 2013, Shinaoka et al. 2020; Shinaoka et al 2024), injecting a contrast agent in the area of lymphatic drainage would lead the agent itself to follow the route that any other substance in the interstitial of tissues would follow, including uptake by lymphatic vessels.

Shinaoka A. (2024). A new lymphography protocol and interpretation principles based on functional lymphatic anatomy in lower limb lymphedema. Anatomical science international, 99(2), 153–158. https://doi.org/10.1007/s12565-023-00754-2

I question the terminology of interrupted lymphatic pathways. Were these pathways really interrupted? Could it be reflux into another connected branching network of lymphatic vessels? Are there other possibilities--- please elaborate. Maybe there could be a statement for the value of histopathology to help figure it out?

The term was firstly used by us because, in our clinical experience, both in dogs and cats, as happened in this cadaveric study, we have occasionally identified a fluorescent lymphatic network that does not appear to drain into any lymph node. Given the exploratory nature of the study, we considered it appropriate to document all findings observed during the lymphographic procedures, describing all the macroscopical features detected. While the possibility of lymphatic reflux or alternative branching pathways was not definitively excluded, our surgical and imaging explorations were designed to detect such phenomena wherever feasible. Further studies will be necessary to confirm or rule out these mechanisms. For this reason, the authors preferred to limit to what was macroscopically detected. While histopathological analyses were considered, the quality of the samples, combined with the fact that the vast majority of cat cadavers were frozen, made a reliable in-depth analysis of fine details not feasible in this experimental setting, and beyond the scope of the present study. Nevertheless, we acknowledge that histopathology could provide crucial insights into the processes behind these observations, and future studies should prioritize obtaining fresh, high-quality samples to facilitate such analyses. Additionally, the territory requiring histopathological examination would have been too large in this context. Future studies focusing on smaller, well-defined areas of interest could incorporate histopathological analyses to provide more precise insights. We agree with the reviewer that further studies are needed to confirm the presence and nature of these drainages and to provide a more detailed characterization. Hence, in the present work, the term “interrupted” is only based on the macroscopic appearance of those vessels and has nothing to do with the histologic or paraphysiological process behind it, which is still unknown.

To clarify this in the text, we modified the phrase as follows (lines 195-198): “- interrupted lymphatic pathways: their definition was used by the authors with a pure descriptive intent and was based only on visible macroscopical evidence, they appeared as fluorescent pathways departing from the injected selected cutaneous region and not arriving at any LC. To confirm the interruption of visible lymphatic pathways, the sites where drainages stopped and the expected and/or closest regional LCs were always surgically explored under NIRF-camera guidance to confirm the absence of fluorescence.”

It is commendable that the cat is the subject of this study as this species is understudied in the literature and is perhaps more common/numerous than dogs in real life. This study may influence clinical decision making for cancer surgery in cats, at least with the awareness of need for a larger anatomic geography when sentinel lymph node mapping is employed.

Thank you for the comment. We are convinced that, nowadays, the feline species deserves increased attention and research. The authors support and strongly believe that the use of cadavers represents a valuable and ethically sound approach in the context of scientific research. We are equally convinced that studies like this can serve as a stimulus for further investigations, both in the field of anatomy and as a potential basis for clinical inferences.

---

## [Decision Letter · Decision Letter 1]

Indocyanine green fluorescence lymphography: an exploratory study of superficial lymphatic territories in the head and hind limbs of 33 cat cadavers

PONE-D-25-09521R1

Dear Dr. Ubiali,

We’re pleased to inform you that your manuscript has been judged scientifically suitable for publication and will be formally accepted for publication once it meets all outstanding technical requirements.

Kind regards,

Douglas H. Thamm, V.M.D.

Academic Editor

PLOS ONE

Additional Editor Comments (optional):

Reviewers' comments:

Reviewer's Responses to Questions

**Comments to the Author**

Reviewer #1: All comments have been addressed

2. Is the manuscript technically sound, and do the data support the conclusions?

Reviewer #1: (No Response)

3. Has the statistical analysis been performed appropriately and rigorously?

Reviewer #1: (No Response)

4. Have the authors made all data underlying the findings in their manuscript fully available?

Reviewer #1: (No Response)

5. Is the manuscript presented in an intelligible fashion and written in standard English?

Reviewer #1: (No Response)

Reviewer #1: (No Response)

**Do you want your identity to be public for this peer review?** For information about this choice, including consent withdrawal, please see our Privacy Policy

Reviewer #1: No

---

## [Editor Report · Acceptance letter]

PONE-D-25-09521R1

PLOS ONE

Dear Dr. Ubiali,

I'm pleased to inform you that your manuscript has been deemed suitable for publication in PLOS ONE. Congratulations! Your manuscript is now being handed over to our production team.

Kind regards,

on behalf of

Dr. Douglas H. Thamm

Academic Editor

PLOS ONE